# Transient Confinement of the Quaternary Tetramethylammonium Tetrafluoroborate Salt in Nylon 6,6 Fibres: Structural Developments for High Performance Properties

**DOI:** 10.3390/ma14112938

**Published:** 2021-05-29

**Authors:** Ahmed Dawelbeit, Muhuo Yu

**Affiliations:** State Key Laboratory for Modification of Chemical Fibers and Polymer Materials, College of Materials Science and Engineering, Donghua University, Shanghai 201620, China; 413004@mail.dhu.edu.cn

**Keywords:** nylon 6,6 fibres, polyamide fibres, crystal orientation, morphological structural developments, quaternary tetramethylammonium tetrafluoroborate, hydrogen bonds

## Abstract

A temporary confinement of the quaternary tetramethylammonium tetrafluoroborate (TMA BF_4_) salt among polyamide molecules has been used for the preparation of aliphatic polyamide nylon 6,6 fibres with high-modulus and high-strength properties. In this method, the suppression or the weakening of the hydrogen bonds between the nylon 6,6 segments has been applied during the conventional low-speed melt spinning process. Thereafter, after the complete hot-drawing stage, the quaternary ammonium salt is fully extracted from the drawn 3 wt.% salt-confined fibres and the nascent fibres are, subsequently, thermally stabilized. The structural developments that are acquired in the confined-nylon 6,6 fibres are ascribed to the developments of the overall fibres’ properties due to the confinement process. Surprisingly, unlike the neat nylon 6,6 fibres, the X-ray diffraction (XRD) patterns of the as-spun salt-confined fibres have shown diminishing of the (110)/(010) diffraction plane that obtained pseudohexagonal-like β’ structural phase. Moreover, the β’ pseudohexagonal-like to α triclinic phase transitions took-place due to the hot-drawing stage (draw-induced phase transitions). Interestingly, the hot-drawing of the as-spun salt-confined nylon 6,6 fibres achieved the same maximum draw ratio of 5.5 at all of the drawing temperatures of 120, 140 and 160 °C. The developments that happened produced the improved values of 43.32 cN/dtex for the tensile-modulus and 6.99 cN/dtex for the tensile-strength of the reverted fibres. The influences of the TMA BF_4_ salt on the structural developments of the crystal orientations, on the morphological structures and on the improvements of the tensile properties of the nylon 6,6 fibres have been intensively studied.

## 1. Introduction

The well-known routes for obtaining high-moduli and high-strength polymeric fibres are the extensibility-based strengthening of the aliphatic molecules as well as the unfoldable-molecules-based stiffening of the aromatic molecules (the stiff molecules do not fold up into lamellae in the presence of aromatic rings) [1]. The extensibility-based strengthening arose, theoretically, from having high deformability along the molecular chains of the semicrystalline fibres [2]. In this aspect, many attempts had been made to improve the tensile properties of polymeric fibres, as a result of the structural developments along the fibres’ axis [3]—such as the flexible ultra-high molecular weight polyethylene [4,5]. However, the aliphatic polyamide nylon 6,6 [6] are the first synthetic fibres that are generally melt spun fibres [7]—which is in contrast to polyethylene, consisting of amide linkages (–NH–CO–) attached directly to two aliphatic groups (alkyl groups) which leads to hydrogen bonded semi-crystalline flexible molecules fibres [8]. On the other hand, the nature of the crystal structure for extruded nylon 6,6 fibres [9] and films [10] is a pseudohexagonal β crystal phase structure that develops into a triclinic α crystal phase structure upon the subsequent cooling process [11] or under water treatment [12]. The nylon 6,6 fibres, since their discovery, have been very important textile and technical fibres [13]. However, the achieved values of the tensile properties still are far away from the theoretical ones [14].

Naturally, the formation of the high-moduli and high-strength nylon 6,6 fibre is, mainly, dependent on the crystalline structure of the as-spun β phases and their development (patently, β to α phases transitions). Here, the hydrogen bonds between the nylon 6,6 molecules are, basically, immobile up to the melting temperature, and these hydrogen bonds constitute the main obstacle towards obtaining the abovementioned high draw ratios and high crystal orientations of the nylon 6,6 fibres. (N.B. one must be careful to differentiate between the as-spun pseudohexagonal β crystal phase and the Brill temperature pseudohexagonal β crystal phase. The former is the crystal structure of the nascent as-just spun fibres (melt extrudate) before the spinline-crystallization and the latter is the structural deformation after the thermal treatment up to the melting point—that causes merging between the intrasheet/interchain (100) and the intersheet (110)/(010) diffraction planes to form a single diffraction at a temperature near 160 °C, i.e., Brill transition [15]. It is to be noted that solution wet spinning only forms fibres that crystallize in triclinic α crystal phase structures [11]). Moreover, in the melt spinning process, nylon 6,6 has fast crystallization kinetics since the formed-yet fibres (the extrudate) tend to crystallize in the spin line in the form of α triclinic crystal phase [16] before the alignment of the molecules! Therefore, to enhance the drawability of the polyamide molecules it is necessary to prevent the crystallization by the suppression or the weakening of the formed hydrogen bonds [17,18] to allow the fibres to be drawn at high draw ratios (plateau deformation), i.e., the highest improvement for the mechanical properties of the fibres will be obtained by the ultimate crystalline orientations [19].

Numerous research methods used to improve the mechanical properties through the structural development of the crystal orientations of nylon 6,6 fibres have been reported, these include spinning (the high-speed melt spinning [20,21,22,23], the wet spinning [24], the dry-jet wet spinning of the reversible solvent-counter-ions (GaCl_3_) complexation [25,26,27,28,29,30,31] and the solution spinning of the reversible superheated water-counter-ions (LiI) shielding [32]) and the drawing (the heat drawing and the annealing [33,34,35,36,37]) methods. It is worth to note here that the montmorillonite clay (MMT) [38,39,40], the multi-wall carbon nanotube (MWNT) [41,42] and the nanodiamond [43] nanocomposite methods as well as the lithium chloride (LiCl) [44] and the ferric chloride (FeCl_3_) [45] interactions have, also, been used to modulate the crystal structure of nylon 6,6 materials.

The quaternary tetramethylammonium tetrafluoroborate (TMA BF_4_) salt is, basically consisting of tetramethylammonium (organic) cations and tetrafluoroborate (inorganic) anions, and has the chemical formula [(CH_3_)_4_N]^+^ [BF_4_]^−^ [46]. Interestingly, its most desirable characteristic is the small sizes (ionic radius) of both the cations and the anions of the TMA BF_4_ salt. The size of the TMA cation is 0.283 nm—which is smaller than the smallest ionic radius of any other alkylammonium cations, where the size of the BF_4_ anion is 0.229 nm [47]. One may note here that this small size makes TMA BF_4_ favourable for many applications. It is widely used in electrochemistry for increasing the conductivity of electrodes [48,49,50,51] or in the carbon electrodes’ voids to enhance the capacitance [52]. Beside their uses in electrochemical applications, they can be confined in supramolecular capsules [53] and deep cavitands [54] as well as intercalate in MMT clay for clay modifications [55]. Additionally, TMA BF_4_ is thermally and electrochemically stable and has good mobility. The nanostructure size of the TMA BF_4_ is smaller than the hydrogen bond distance between two adjacent molecules [56]. TMA BF_4_ is also a water-soluble material with a high-degree of inter- and intra-molecular hydrogen bonding.

In this research work, novel nylon 6,6 fibres are prepared by the temporal reversible salt-confined method using low-speed melt spinning, so as to produce high-strength and high-modulus fibres. The nylon 6,6 resin and TMA BF_4_ salt are melt spun by a single-screw melt spinning machine followed by a hot-drawing process (for the as-spun salt-confined fibres) with different drawing ratios and at different drawing temperatures. Then, subsequently, salt-extraction processes followed by immediate thermal stabilization processes are applied to the fibres for the reversion of the pristine ones (N.B.: The reversion processes of the fibres are applied when the molecular chains are totally oriented and are fully extended).

## 2. Materials and Methods

### 2.1. Materials

The aliphatic polyamide polymer pellets nylon 6,6 Zytel^®^ 101L—used here—is manufactured by The DuPont Company Ltd., (Wilmington, NC, USA), while, tetramethylammonium tetrafluoroborate electrolyte salt (CH_3_)_4_N(BF_4_) is purchased from The Shanghai ChengJie Chemical Company (Shanghai, China).

### 2.2. Fibre Formation

The spinning and the take-up machines are manufactured by ABE Engineering Company (Tokyo, Japan), while the parallel drawing machine which used for hot-drawing is manufactured by Suzhou Electrical Technology Development Company (Suzhou, China). In the confinement process of the nylon 6,6 fibres—made here—the aliphatic polyamide nylon 6,6 pellets and the TMA BF_4_ salt are co-fed into a single screw melt spinning machine (Temtec, Kyoto, Japan) at temperatures in the 295–300 °C range. The salt weight to nylon 6,6 weight ratios are 0, 1, 2, 3 and 4%. The extrudate is spun out through a spinneret of 36 holes with a diameter of 0.3 mm for each hole. However, the TMA BF_4_ salt-confined nylon 6,6 fibres (thereafter is referred to as salt-confined nylon 6,6 fibres (the abbreviation “IS” in the figures’ captions and the tables’ entries, also stands for TMA BF_4_ salt) have been wound up at a winding speed of 306 m/min for neat and salt-confined nylon 6,6 fibres. Moreover, all of the as-spun fibres are individually subjected to hot-drawing processes at temperatures of 120, 140 and 160 °C. The results of the melt spinning and the hot-drawing processes of the salt-confined nylon 6,6 fibres revealed—experimentally—that, the additional amount of the TMA BF_4_ contents influence the maximum achievable drawing ratios of the salt-confined fibres—which is very low for salt contents of less than 3%. In this piece of work, out of the total of the nylon 6,6 salt-confined fibres which are prepared, only those that are confined with 3 wt.% TMA BF4 and hot drawn at the temperatures of 120, 140 and 160 °C are chosen to test whether the salt-confined at low-speed melt spinning method is worked well for improving the mechanical properties of the nylon 6,6 fibres, or not-concentred studies has been made on theses fibres because they have shown the greatest drawing ratio (of 5.5) and, consequently, the greatest extensibility.

The salt-extraction process has been carried out by immersing the drawn salt-confined nylon 6,6 fibres into a 10% pentahydrate sodium thiosulphate (Na_2_S_2_O_3_·5H_2_O) solution at room temperature for about 12 h. The treated fibres are, subsequently, washed with a deionized water for another hour and, finally, the resulting fibres are dried in an oven at 50 °C for one hour. Thereafter, the salt-free regenerated fibres—referred to here as the regenerated fibres—have been subjected to a thermal stabilization treatment in a heating tube at 190 °C under tension in an inert nitrogen gas atmosphere, as illustrated in Scheme S3 below. The resulted thermally stabilized fibres are referred to as the stabilized fibres.

### 2.3. Characterization Techniques

#### 2.3.1. Fourier Transforms Infra-Red Spectroscopy (FTIR)

A Fourier-transform Infra-red (FTIR) spectrometer is used to characterize the chemical composition and to differentiate between the crystalline structure phases of the nylon 6,6 fibres and their development due to the deformations processes. The crystal structure developments of the salt-confined nylon 6,6 fibre have been investigated using a Nicolet 6700 FTIR-ATR instrument (Thermo Fisher Scientific Inc., Waltham, MA, USA). The scanning number for each sample is 16 and 0.482 cm^−1^ for the data spacing used. The data fitting and the analyses are processed by the OMNIC (Thermo Fisher Scientific Inc., Waltham, MA, USA) and the Origin Lab (8.5, Origin Lab corporation Northampton, MA, USA) softwares. 

#### 2.3.2. X-ray Diffraction (XRD)

The crystal structures of the neat and the salt-confined nylon 6,6 fibres are investigated using an X-ray polycrystal diffractometer (XRD)—D/max-2550 PC, manufactured by Rigaku (Tokyo, Japan).

##### Crystal Size

The apparent crystallite size (*ACS*) of the neat and the salt-confined nylon 6,6 fibres have been calculated by the Scherrer equation (Equation (1)):(1)$$ACS=\frac{K\lambda }{\beta \cos\theta }$$
where: *θ* is the Bragg diffraction angle of the equatorial plane, *λ* (=1.541 Å) is the X-ray wave length and *K* is a constant whose value is dependent on the crystallite shape.

##### Crystal Orientation

The orientation parameters $\langle cos^{2}\emptyset_{hkl}\rangle$ are calculated from the mean-square cosine of the angle ($\emptyset_{hkl}$) and are defined by the equation (Equation (2)):(2)$$\langle cos^{2}\emptyset_{hkl}\rangle =\frac{\int_{0}^{\frac{\pi }{2}}I\left(\emptyset_{hkl}\right)cos^{2}\emptyset_{hkl}\sin\emptyset_{hkl}d\emptyset_{hkl}}{\int_{0}^{\frac{\pi }{2}}I\left(\emptyset_{hkl}\right)\sin\emptyset_{hkl}d\emptyset_{hkl}}$$
where $\emptyset_{hkl}$ is the azimuthal angle measured from the equatorial planes and $I\left(\emptyset_{hkl}\right)$ is the fully corrected intensity distribution. The values of the orientation parameter $\langle cos^{2}\emptyset_{hkl}\rangle$ of the molecules have the values of unity for the parallel oriented molecules, zero for perpendicularly oriented molecules and one third for randomly oriented molecules to the fibre axis.

The Hermans’s orientation function ($f_{c}$) describes the degree of orientation via the equation (Equation (3)):(3)$$f_{c}=\frac{\langle cos^{2}\emptyset_{hkl}\rangle -1}{2}$$
where the angle ($\emptyset_{hkl}$) is the angle between the polymer chains and the fibres axis.

The Hermans’s orientation function ($f_{c}$) can be seen to have the values of unity when the molecular chain of the fibres is paralleled with the fibre axis, −0.5 when the molecular chain is perpendicular to the fibre axis and zero when the molecular chain is totally unoriented.

##### The Crystal Perfection Index (CPI)

The crystal perfection index (CPI) for the neat and the salt-confined nylon 6,6 fibres is defined by the equation (Equation (4)):(4)$$CPI\%=100\times \left(\frac{\left(d_{100}/d_{010}\right)-1}{0.189}\right)$$
where *d*_100_ and *d*_010_ are the interplanar lattice spacings for the (100) and the (010) diffraction planes, respectively, and the factor 0.189 is the corresponding value for the well-crystallized sample (the ideal crystal structure).

#### 2.3.3. Nuclear Magnetic Resonance (NMR)

The solid state ^13^C-NMR measurements of the as-spun neat and salt-confined nylon 6,6 fibres have been carried out on an AVANCE 400 NMR spectrometer (Bruker, Switzerland). The fibres are cut into short pieces, prior to the investigation, and filled into a cylindrical ceramic rotor for ^13^C CP/MAS spectra, The NMR spectrometer was operated at 100 MHz for the samples and spun at spin rate of 5 kHz. The glycine upfield peak (176.03 ppm) relative to tetramethylsilane (TMS) [(CH_3_)_4_ Si] is used for the calibration.

#### 2.3.4. Scanning Electron Microscope (SEM)

The morphological structures of the neat and the salt-confined nylon 6,6 fibres are obtained using a Field Scanning Electron Microscope (SEM)—Hitachi SU8010 and S-4800—supplied by Hitachi (Tokyo, Japan).

#### 2.3.5. Mechanical Properties

The mechanical properties of the neat and the salt-confined nylon 6,6 fibres were determined by a XL-2 tensile strength testing machine for fibre bundles. The machine is supplied by Shanghai New Fibre Instrument Limited Company (Shanghai, China). A pneumatic device is used for clamping the fibres during the tensile measurement, with the effective gauge length set of 50 mm. The mechanical properties of the reverted fibres are characterized by an XL-1 Tensile Strength Testing Machine also supplied by Shanghai New Fibre Instrument Limited Company, Shanghai—with clamping of 10 mm for each single fibre. A pneumatic device id also used for clamping the fibres during the tensile property measurements.

## 3. Results and Discussions

### 3.1. Structural Development of the Salt-Confined Nylon 6,6 Fibres

#### 3.1.1. Fourier Transform Infra-Red Spectroscopy (FTIR)

FTIR spectroscopy is used to investigate the chemical and the crystal structure of the quaternary ammonium salt, the as-spun and drawn neat nylon 6,6 fibres and the as-spun and drawn salt-confined nylon 6,6 fibres with different contents. The bands assignments of the FTIR spectra in the ranges of 4000–600 cm^−1^, 1500–1100 cm^−1^ and 1100–800 cm^−1^ for the as-spun neat and for the as-spun salt-confined nylon 6,6 fibres are illustrated in Figure 1a–c. The FTIR results show that all of the peaks of the neat nylon 6,6 fibres also occur in the salt-confined nylon 6,6 fibres [57]. The FTIR spectra for both of the neat and the salt-confined nylon 6,6 fibres show peaks at 936 and 906 cm^−1^ which are attributed to the crystalline phase structure for both types of the fibres. The FTIR spectra for the neat fibres show a peak at 1640 cm^−1^ attributed to the fibres’ amorphous phase structure. As for the salt-confined fibres, the FTIR spectra show for them peaks in the range 1639.26–1639.70 cm^−1^. However, the FTIR peak at 1180 cm^−1^ for the neat fibres does not change its position due to either the salt-confinement or due to the hot drawing processes—this peak is used as a reference band for the nylon 6,6 fibres. This observation of assigning the 1180 cm^−1^ band as a reference band for the nylon 6,6 fibres was also mentioned by Vasanthan and Salem [58] for the FTIR band assignment of the heat-treated nylon 6,6 fibres as can be deduced from Figure 2a.

It is worth noting that, contrary to Vasanthan et al. [58,59], the above FTIR results found in this research study do not show any crystalline phase peaks at 924 cm^−1^ for any of the types of the nylon 6,6 fibres considered.

On the other hand, the FTIR spectrum bands of the TMA BF_4_ salt are, mainly classified based on the tetragonal symmetry of the arrangement of the methylene groups in the [(CH_3_)_4_N] cation and the disordered [BF_4_] anion which located with the one B–F bond lying on the fourfold (tetragonal) axis and the other three F-ions making a triangle with a disordered distribution (trigonal axis) [49,60]. The FTIR results show absorption bands at 949, 1292, 1410, 1492 and 3052 cm^−1^ that are attributed to the [(CH_3_)_4_N] cations and, also, bands at 769, 1034 and 1049 cm^−1^ for the [BF_4_] anions [61,62]. The band at 3052 cm^−1^ is interpreted to be due to the N–CH_3_ stretching mode and the symmetric and antisymmetric C–H bending of the methyl terminal belonging to the cations. The band at about 949 cm^−1^ is found to be compatible with the breathing of the NC4 skeleton, although that band at 1049 cm^−1^ can be assigned to the stretch vibration modes of the [BF_4_] anions groups too.

Moreover, the FTIR spectra showed four new absorption bands for the 3 wt.% as-spun salt-confined nylon 6,6 fibres at the wavenumbers 1494 cm^−1^, 1054 cm^−1^, 1041 cm^−1^ and 951 cm^−1^. These bands, which can clearly be observed in Figure 2b, are attributed to the TMA BF_4_—which showed a red shift for the stretching vibrations [63] away from that for the pure TMA BF_4_ salt. This result is believed to occur as a result of the hydrogen bonding interactions between the cations and the anions of the TMA BF_4_ salt and the carbonyl (donors) and the amine (acceptors) segments of the amide groups that took-place among the nylon 6,6 molecules. Despite the hydrogen bonds interactions between the polyamide segment are fixed up to the melting temperature and also did not change with molten salt [64,65], the as-spun salt-confined fibres show that the full width at half maximum (FWHM) of the carbonyl peak of the solid state ^13^C-NMR is wider than that of the neat nylon 6,6 fibres—which is an evidence for the interruption of the hydrogen bonds through the salt-confinement process—see Figure 3. Furthermore, the absorption band at 936 cm^−1^ of the as-spun salt-confined nylon 6,6 fibres showed a slight shift to the smaller values of 934 cm^−1^ and 935 cm^−1^ due to the hot-drawing stage for the drawing ratios of 4.5, 5.0 and 5.5 at the drawing temperatures of 120, 140 and 160 °C (see Appendix A).

#### 3.1.2. X-ray Diffraction (XRD)

Generally, in the formation of the semicrystalline nylon 6,6 fibres, the relationships between the undrawn fibre structure, the drawing conditions, the structures and properties of the drawn fibres are considered to be essential features to improve the drawing process for achieving high tensile strengths and high tensile moduli for the nylon 6,6 fibres. Here, it is known that, the drawing conditions, the structures and properties of the resulting drawn fibres (triclinic α crystal phase structure) are strongly governed by the quality and the characteristics of the original-undrawn as-spun fibres (pseudohexagonal β crystal phase structure). The molecular extensibility and the crystallites alignment along the fibre axis of the amide molecules (molecular orientations) are regularly concomitant with the changes in the phase structure (β to α phases transitions) and in the other morphological structure features (spherulites and fibrillar structures).

##### The Crystal Structure

The XRD diffraction patterns of the as-spun neat and the salt-confined nylon 6,6 fibres are shown in Figure 4 and Figure 5 and the values of the crystal parameters are tabulated in Table 1. The results shown here indicate that the hydrogen bonding among the molecular segments of the as-spun nylon 6,6 molecules has been suppressed or interrupted by the confinement of the TMA BF_4_ inorganic salt.

It is known that the crystallization of the just spun molecules (as-spun fibres) adopts a staggered arrangement of the β crystal phase structure yielding a pseudohexagonal unit cell. The structural developments at the formation of the as-spun nylon 6,6 fibres yield a stable α crystal phase structure. The resulting α crystal phase exhibits progressive arrangements of the amide molecules with a triclinic unit cell structure [66,67]. The TMA BF_4_ salt has a monoclinic unit cell [68] and the highest diffraction peak occurred at two-theta of 21.5° with the interplanar spacing of 4.126 Å—as shown in Appendix A. While, the results of the TMA BF_4_ salt confinement among the molecular chains of the nylon 6,6 fibres show an interruption of the hydrogen bonds between the carbonyl and the amine segments of the amide functional group—which turn in inhibiting the melt crystallization of the salt-confined nylon 6,6 fibres. This is an indication that weakening took place among the hydrogen bonds (i.e., the effectiveness of the hydrogen bonds has been reduced) in the salt-confined fibres. Surprisingly, at the drawing process, these results lead to a maximum draw ratio of 5.5 for the 3 wt.% salt-confined nylon 6,6 fibre for all of the drawing temperatures of 120, 140 and 160 °C. On the other hand, the maximum draw ratio of the neat nylon 6,6 fibres varies from 5.0 at the drawing temperatures of 120 and 140 °C to 5.5 at the drawing temperature of 160 °C.

Interestingly, the salt-confined nylon 6,6 fibres reveal that, the β’-to-α phase transitions took-place during the hot-drawing stage (stress-induced phase transitions) to develop a triclinic α crystal phase structure—as shown in Figure 5, Appendix A and Appendix A. This finding is contrary to the results described by Ramesh et al. [11] and Vergelati et al. [12].

The interplanar spacing (d-spacing) of the (100) and (110)/(010) diffraction planes takes values of 4.3671 and 3.8541 Å, respectively, for the as-spun neat fibres and the spacing values of 4.3763 and 3.8854 Å respectively, for the drawn neat fibres at drawing temperatures of 140 °C. In the confined fibres, however, the XRD pattern of the as-spun 3% salt-confined fibres rises two diffraction peaks for the (100) and for the (110)/(010) diffraction planes at 20.550° and 23.132° with the interplanar spacing of 4.3184 and 3.8419 Å, respectively. Furthermore, the drawn salt-confined fibres at drawing ratio of 5.5 and a drawing temperature of 140 °C exhibits two diffraction peaks at 20.298° and 23.110° with the interplanar spacings of 4.3713 and 3.8455 Å, respectively.

Moreover, the salt confinement among the molecular chains of the nylon 6,6 fibres led to the variation in the interplanar spacing values for the as-spun neat and salt-confined fibres that is reflected in developing small crystal sizes—as shown in Table 2. It is to be noted here that the position of the XRD diffraction peak for the (100) plane is increased when the drawing temperature is raised from 140 to 160 °C at the drawing ratio of 5.5—this can be attributed to the structural deformations at the Brill transition temperature (T_B_) of the nylon 6,6 [69].

##### Crystal Orientation

Typical drawing of the semicrystalline nylon 6,6 fibres occurs by pulling out the polyamide molecules from the folded lamellar crystal into well oriented crystals, without losses (breaking) in the taut molecules and without losses in the covalent bonded chain (backbone molecule). The as-spun neat and the as-spun 3% salt-confined nylon 6,6 fibres exhibit unoriented molecular orientations of the polyamide molecules. However, the molecular orientations of the neat and the salt-confined nylon 6,6 fibres greatly increase with the increase of the drawing ratios from 1.0 to 5.5. These results revealed that the salt confinement of the ammonium salt among the nylon 6,6 molecules have strongly affected the existed hydrogen bonding at the melt spinning process and have inhibited the crystallization of the molten extrudate of the nascent nylon 6,6 fibres. Indeed, the full width at half maximum (FWHM) has, clearly, showed random crystal orientations of the as-spun confined fibres in comparison with the neat ones—as seen in Figure 6. The FWHM of the azimuthal angle decreases due to the increase of the deformation when both types of the as-spun nylon 6,6 fibres are hot-drawn. Moreover, the effect of changing the drawing temperature on the Herman’s orientation function of the as-spun and drawn neat fibres and the as-spun and drawn salt-confined nylon 6,6 fibres takes different values at the same fixed drawing ratio. Furthermore, the Herman’s orientation function has exhibited the highest value at the drawing ratio 5.5 at the drawing temperature of 140 °C. 

These fibres have, also, obtained the smallest value for the orientation angle (the angle between the polymeric molecular chains and the fibre axis)—as shown in Figure 7. Surprisingly, increasing the drawing temperature has no effect on the extensibility (draw ratio).

##### Crystallinity

The as-spun neat nylon 6,6 fibres exhibit two peaks for the interchain/interasheet (100) and the intersheet (110)/(010) diffraction planes, but, in contrast to the neat fibres, the as-spun salt-confined nylon 6,6 fibres show for the (100) peak relatively high intensity compared to the intensity of the intersheet (110)/(010) peak due to the interaction of the TMA BF_4_ salt with the amide group via the hydrogen bonds. This observation confirms the breaking of the hydrogen bonds—as shown in Figure 8. The results, here, give values of the crystallinity of the as-spun salt-confined fibres higher than that of the as-spun neat fibres—which are melt-spun at the same melting temperatures and at the same spinning speed conditions. The increase of the crystallinity of the as-spun salt-confined fibres is believed to be attributed to the effect of the crystal size of the nascent fibres. Moreover, the small crystal size of the salt-confined crystals can be ascribed to the influences of the nanoscale size of the TMA BF_4_ salt—which have the ability to infiltrate inside the lamella structure of the spherulites. However, when the drawing temperature is increased to 140 °C the value of the crystallinity for the salt-confined fibres has exhibited a slight reduction for the high drawing ratio of 5.5—see Table 2. Here, the crystallinity values for the drawn fibres had been reduced to values lower than that for the as-spun salt-confined ones—which is attributed to morphological structurals’ changes due to the partial breaking of the amide (original) hydrogen bonding and forming strong (new) hydrogen bonds between the TMA BF_4_ salt and the amide groups. In addition, the intermediate structures (mesophases) tend to transform into amorphous phases *not* crystalline ones [70].

##### The Apparent Crystal Size (ACS) and The Crystal Perfection Index (CPI)

It worth noting that, due to the salt confinement of the TMA BF_4_ salt, the as-spun salt-confined fibres exhibited crystal sizes (ACS) smaller than that of the as-spun neat fibres—see Figure 9. The crystal size of the as-spun 3% wt salt-confined fibres had been found to be smaller than half the crystal size of the as-spun neat nylon 6,6 fibres in both the hydrogen bonded (interasheet/interchain) and the van der Waal (intersheet) directions. A possible explanation is that the salt confinement lead to hindering the large crystal growth. This is similar to the nanotube net-work structure of the MWNT-modified nylon 6,6 nanocomposites [41]. However, when applying the hot drawing process, the ACS of the neat drawn fibres at the (100) and (110/010) planes concurrently decreases with the increase of the drawing ratio as shown in Appendix A. On the contrary, it has been observed that the converse takes place for the salt-confined nylon 6,6 fibres i.e., its crystal size increases with increasing the drawing ratio—see Appendix A. This result revealed that the crystals sizes variations of the (110/010) planes during the drawing processes at different drawing temperatures—for the salt-confined fibres—are much smaller than the crystal size of the (100) plane—where the hydrogen bonding takes-place and the crystal growth is fast. However, the ACS_100_ show significant increase, at the same drawing ratio 5.5, with the increase of the drawing temperature. Obviously, here, the temperature has more effect than the strain. This observation is in agreement with the data reported by Hsiao et al. [69].

On the other hand, the as-spun 3% salt-confined fibres have shown a crystal perfection index (CPI) of lower value than that for the neat nylon 6,6 fibres. This observation may be attributed to the interaction of the TMA BF_4_ salt and the nylon 6,6 among the polyamide molecules which led to the variations in the interplanar spacings (Bragg’s spacings)—as mentioned above. Moreover, the drawn salt-confined nylon 6,6 fibres have shown considerable improvements in the values of the CPI during the drawing process for all of the drawing ratios and for all of the drawing temperatures. The values of the CPI of the as-spun and drawn 3% salt-confined nylon 6,6 fibres are plotted in Figure 10, below.

#### 3.1.3. Scanning Electron Microscopy (SEM)

The morphological structures of the surfaces and the cross-sectional areas of the as-spun and drawn salt-confined nylon 6,6 fibres are observed by means of the SEM technique. The SEM micrographs obviously show the axialite or hedrite morphological structure (structure visualized as two hexagonally shaped placed spine to spine) on the surface of the as-spun salt-confined fibres—which eventually develops into spherulites structures [59,71]. However, when the fibres are drawn, these spherulites will turn into the developing of the fibrillar structure. These fibrillar structures have appeared as parallel streaks throughout the surfaces of the fibres as shown in Figure 11. This behaviour has, similarly, been observed during the drawing process of the semicrystalline polyethylene and polypropylene polymers [59]. Besides the molecular orientations and the hydrogen bonds, the microfibrillar morphology determines the mechanical properties of the nylon 6,6 fibres, as well [72]. Moreover, the SEM micrograph, also, has illustrated the TMA BF_4_ particles on the surfaces of the fibres. Nevertheless, the cross-sectional views have shown some void structures for both of the as-spun and the drawn fibres. The cross-sectional areas of the drawn salt-confined fibres have shown a pleated-sheet structures as a result of the improvements of the spherical structures due to the hot drawing processes.

### 3.2. Mechanical Properties

Tentative secession of the hydrogen bonds by means of temporary confinement processes have been used for the preparation of high-modulus and high-strength nylon 6,6 fibres (the high-modulus and the high-strength fibres means the high-performance aliphatic nylon 6,6 fibres). The tensile mechanical properties of the neat and the salt-confined nylon 6,6 fibre are tabulated in Table 3. The results have revealed that, both of the initial moduli and the tensile strengths of the salt-confined nylon 6,6 fibres increase with increasing of the draw ratio—as shown in Table 3 and Appendix A. The tensile moduli values for all of the salt-confined fibres are higher than that for the neat nylon 6,6 fibres at all of the drawing ratios and at all of the drawing temperatures. However, at the drawing temperatures of 120 and 140 °C, the values of the tensile strength for the salt-confined fibres at their maximum drawing ratio (of 5.5) are higher than that for the neat fibres at their maximum drawing ratio (of 5.0). On the other hand, the drawn salt-confined fibres at the drawing temperature 160 °C and its maximum drawing ratio of 5.5 have shown a slight reduction in the tensile strength value as shown in Figure 12—this can be ascribed to the structural deformations at the Brill transition temperature (T_B_) of nylon 6,6 [15]. Moreover, the values of the elongations for all the salt-confined fibres are lower than that values for the neat nylon 6,6 fibres for all of the drawing ratios and at all the drawing temperatures- as a result of the crystal orientation development of the nylon 6,6 molecules.

### 3.3. The Extraction of the Confined TMA BF_4_ Salt and the Thermal Stabilization Processes

The reversion process has been made after studying the structural and the mechanical properties of the TMA BF_4_ salt-confined nylon 6,6 fibres by the FTIR, SEM, the XRD techniques and mechanical properties.

The FTIR of the salt-confined, regenerated and stabilized nylon 6,6 fibres, as shown in Figure 13a–c, showed the removal of the confined salt from the 3% salt-confined nylon 6,6 fibres—which was drawn at the maximum obtainable drawing ratio of 5.5 and at drawing temperatures of 120, 140 and 160 °C. However, the FTIR spectra show a slight red shift for all of the reverted fibres (an increase in energy). The decrease in the wavelength of the peaks’ positions of the reverted fibres, emphasizes the hydrogen bond re-formation between the amide donors and acceptors sides.

The XRD patterns have shown that, the peaks’ heights for the diffraction peaks of the regenerated and the stabilized fibres for the (110/010) plane are increased (higher than the peak height of the (100) plane) due to the extraction of the TMA BF_4_ salt and the stabilization of the regenerated fibres. Moreover, the peaks positions of the (100) and the (110) planes are shifted towards higher values. Besides the peaks shifting (positions’ changes), these results of the XRD testing experiments on the reverted fibres show, also, changes in the peaks widths’. It is known that, the (110/010) peak corresponds to the direction of the hydrogen bonding interchain/interasheet. Thus, the reforming of the hydrogen bonding took place among the neighbouring of the extended molecules for the fibres reverted from the drawn 3 wt.% salt-confined nylon 6,6 fibres drawn at the drawing ratio of 5.5 at drawing temperatures of 120, 140 and 160 °C—see Figure 14. Therefore, the crystallinity of the reverted fibres tends to increase due to the crystallization of the well oriented molecules in the amorphous region which engaged in contributing to the crystallinity. The crystal size of the (100) plane of the reverted fibres has exhibited small values—smaller than that values for neat drawn nylon 6,6 fibres (ACS_100_ = 51 Å). On the other hand, the crystal perfection index (CPI) of the regenerated fibres has shown values higher than the corresponding values for the drawn salt-confined fibres at all of the drawing temperatures of 120, 140 and 160 °C. The crystal orientations of the reverted fibres have, also, shown significant improvements due to the reversion processes. The XRD results are tabulated in Appendix A. Scheme 1 illustrates the variations of the orientation factor through all of the salt-confined fibres formation and their reversion to the pure nylon 6,6 fibres.

The SEM micrographs illustrate the morphological structures of the surfaces and cross-sectional areas of the reverted nylon 6,6 fibres. The morphological structures of the regenerated fibres achieved voids structures on their surfaces and their cross-sections. The reverted fibres have shown smooth surfaces—as shown in Figure 15 for the regenerated and the stabilized nylon 6,6 fibres (the regeneration have been done for the 3% salt-confined nylon 6,6 fibres drawn at drawing ratio of 5.5 and drawing temperature of 140 °C). However, the cross-sectional areas for the stabilized fibres that are regenerated from the 3 wt.% salt-confined nylon 6,6 fibres drawn at temperature of 120 °C and a draw ratio of 5.5 have shown folded lamellar structure.

The development of the crystal structure of the melt-spun salt-confined nylon 6,6 fibres is reflected in the improvements of the mechanical properties of the three types of the developed nylon 6,6 fibres (the salt-confined, the regenerated and the thermally stabilized fibres)—as tabulated in Table 3 and Table 4. The results have shown tensile modulus 43.32 cN/dtex and tensile strength of 6.99 cN/dtex for the reverted fibres.

Furthermore, the appearance of the salt-confined and the reverted fibres has reflected silver colour—which is known for the melt spinning nylon 6,6 fibres. This silver colour implies the existence of voids inside the resulting fibres [73]. However, an additional SEM analysis using high magnifications have been carried out to investigate the voids formations. The SEM results have, obviously, confirmed that, the morphological structures of the salt-confined and the reverted nylon 6,6 fibres have some voids and cracks structures on their surfaces as shown in Figure 16, below. This formation of the voids structures can be thought about to be the main obstacle towards achieving great improvements in the mechanical properties of the resulting fibres—despite the fact that the achieved values of the tensile strength of the fibres in the above mention reversible TMA BF_4_ salt confinement method are higher than their corresponding values for the neat nylon 6,6 fibres.

## 4. Conclusions

The application of the reversible method of confinement by quaternary tetramethylammonium tetrafluoroborate (TMA BF_4_) salt that has been applied among the polyamide molecules of the nylon 6,6 fibres to inhibit (or to weaken) the hydrogen bond formation during the melt spinning process has shown great success. The XRD patterns obtained pseudohexagonal-like β’ structural phase for the (110)/(010) diffraction plane of the as-spun salt-confined fibres that have been hot-drawn to orient the molecules along the fibres’ axis draw-induced phase transitions (i.e., β’-to-α phase transitions took-place during the hot-drawing stage). This leads to the formation of relatively high-strength and high-modulus fibres. The salt-confined nylon 6,6 fibres achieved ultimate molecular orientations of 5.5 at all of the (used) drawing temperatures and also, exhibited crystal size (ACS) smaller than that of the neat fibres under the same conditions which have been developed during the drawing process. Moreover, the structural developments lead to the improvements of the crystal orientation, the morphological structures as well as the mechanical properties. Interestingly, increasing the drawing temperature is found to have no effect on the molecular chain extensibility (draw ratio). This observation suggests that the ammonium salt confinement took place in the amorphous phase. The silver–coloured appearance of the salt-confined and the reverted nylon 6,6 fibres is an evidence of the formation of voids structures inside these fibres. Despite of the formation of these voids, the mechanical properties of the reverted fibres are improved. These improvements are reflected in the achievements of the reverted fibres of an improved tensile modulus of 43.32 cN/dtex and an improved tensile strength of 6.99 cN/dtex (these values can be more improved if a way is found to prevent voids formations inside the fibres). Thus, it can be concluded that the application of the reversible salt confinement method to the nylon 6,6 fibres in the melt spinning process leads to the preparation of high-modulus and high-strength polyamide fibres. Moreover, the extraction of the ammonium salt provides a desirable way to achieve a commercially beneficial fibres’ formation process.

## Data Availability

Not applicable.

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
