# Peer review of "Transient Confinement of the Quaternary Tetramethylammonium Tetrafluoroborate Salt in Nylon 6,6 Fibres: Structural Developments for High Performance Properties"

_materials, 2021, doi:10.3390/ma14112938_

Round 1

Reviewer 1 Report

The comments and suggestions concerning the manuscript "Transient Confinement of the Quaternary Tetramethylammonium Tetrafluoroborate Salt in Nylon 6,6 Fibres: Structural Developments for High Performance Properties" (materials-1215537) can be found in the provided attachment.

Author Response

Dear Reviewer

Thank you very much for the effort and time you put into your review of the manuscript “Manuscript ID: materials-1215537” titled “Transient Confinement of the Quaternary Tetramethylammonium Tetrafluoroborate Salt in Nylon 6,6 Fibres: Structural Developments for High Performance Properties".

The comments have been carefully considered and responded. We have incorporated all of the suggestions made by the reviewers. The point-by-point response to the reviewers’ comments and concerns can be seen in the attached file.

Sincerely yours,

Reviewer 2 Report

  1. Please mention explicitly if Fig. 1, Fig. 2 and all others represent personal contribution and authors' research results.

2. Please reconsider paper structure and introduce chapter 2. Materials and methods into the paper (not supplementary material).

3.  Which is the application, for the presented research, of equations (1), (2) and the rest - from the Supplementary material. For the time being, they represent general, theoretical relationship.

4. At Fig. 7, Fig. 8, please define the Drawing ratio

5. At Fig. 12 - mention the vertical axis unit 

6. At lines 154, line 242, line 266 -  correct the unit for Degrees Celsius

7. Reconsider the title of subchapter 2.1 - as it is rather characterization, than  structural development phases

8. Please present images of the how the presented results were obtained 

Author Response

(The authors gave the same response as above.)

Round 2

Reviewer 2 Report

Highly improved paper.